# Environmental Fatigue Behavior of a Z3CN20.09M Stainless Steel in High Temperature Water

Kewei Fang [1,*], Kunjie Luo [1], Li Wang [1], Chengtao Li [1], Lei Wang [2] and Yanxin Qiao [2,*]

[1] Life Management Center, Suzhou Nuclear Power Research Institute, Suzhou 215004, China; luokunjie@cgnpc.com.cn (K.L.); lwongsz@foxmail.com (L.W.); lichengtao@cgnpc.com.cn (C.L.)

[2] School of Materials Science and Engineering, Jiangsu University of Science and Technology, Zhenjiang 212003, China; wangl_ray@just.edu.cn

* Correspondence: fangkewei@cgnpc.com.cn (K.F.); yxqiao@just.edu.cn (Y.Q.)

**Abstract:** The low-cycle fatigue behavior of a Z3CN20.09M austenitic stainless steel was investigated and its fatigue life in high temperature water was compared to that in the air at room temperature. It is found that the fatigue life in water at 300 °C was shorter than that in air, and it decreased with the decreasing strain rate from 0.4% to 0.004%/s. The ductile striations having streamed down features were observed at the strain rate of 0.004%/s, indicating that Z3CN20.09M austenitic stainless steel experienced anodic dissolution. The fatigue life obtained in the present experiment was consistent with that using prediction models.

**Keywords:** Z3CN20.09M; corrosion fatigue; fatigue life; high temperature water





## 1. Introduction

Corrosion and environmental fatigue damage is one common material degradation process in nuclear power plants [1–6]. The wide use of austenitic stainless steels (ASSs) to fabricate nuclear power plant components raises the importance to investigate their low-cycle fatigue (LCF) properties in a light water reactor (LWR) environment to ensure the integrity and safety of nuclear power plants [3,4,6,7]. Furthermore, the ASME design fatigue curve cannot explicitly address the contribution of the service environment to the service life of these ASS components [3,5,8]. Recently, many studies related to corrosion fatigue of SSs have been performed and tried to address the effect of the corrosive environment at high temperatures (i.e., 300 °C) on their fatigue life and possible fatigue mechanism [2,5,6,9]. A large number of research data on corrosion fatigue of ASSs in simulating the environment in LWR were collected by Keisler et al. [2] in the Argonne National Laboratory (ANL). Based on these data, Chopra et al. [3] established a statistical model and can use it to predict the fatigue life of ASSs in LWR environments successfully. Higuchi et al. [5,10] have also evaluated the fatigue life data of SSs reported by Japanese researchers and have proposed a parameter named the fatigue life correction factor ($F_{en}$) to estimate the fatigue life of SSs in LWR environments. Although there are many similarities between the ANL's model and Higuchi's model, differences can also be found in them, such as loading parameters, the working environments, material variability, etc. [8,11].

The Z3CN20.09M ASS is commonly used in PWR nuclear power plants in China. The corrosion fatigue properties of Z3CN20.09M ASS in a simulated LWR environment are closely related to the safe operation of nuclear power plants. In the present work, the LCF tests were conducted to explore the fatigue life of the Z3CN20.09M ASS. The obtained fatigue life of this work was compared with the ANL and Higuchi models. Additionally, the observation of the fractured surface of Z3CN20.09M ASS was conducted to further clarify the corrosion fatigue behaviors in a simulated LWR environment.

## 2. Materials and Experimental Procedures

The test rig used in the present study is composed of an autoclave chamber, a high temperature and pressurized water loop, a data acquisition system, and control equipment, as shown in Figure 1. It simulates the operating conditions in LWR via controlling the content of dissolved oxygen (DO) in water, pH, and water conductivity.

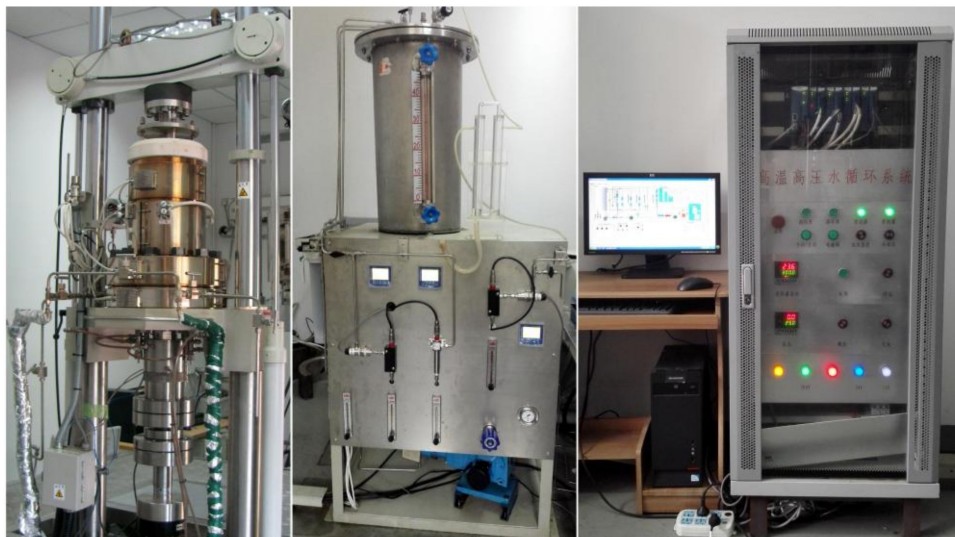

**Figure 1.** An image showing the instrument setup for a low-cycle fatigue test.

The material studied was the Z3CN20.09M ASS used as the primary piping material in nuclear power plants in China. Its chemical composition (in wt.%) was: 0.015% C, 1.08% Si, 1.09% Mn, 0.018% P, 0.002% S, 19.63% Cr, 9.27% Ni, 0.02% Cu, 0.04% Co, and bal. Fe. The mechanical properties of Z3CN20.09M ASS at 25 °C (room temperature, RT) and 350 °C are given in Table 1. Figure 2 shows the microstructure of the Z3CN20.09M ASS, which has a duplex structure: gray δ-ferrite distributing as a network in a white austenitic matrix, which has been characterized carefully by Xue et al. [12].

The LCF test conditions, including strain rate, strain amplitude, load ratio (R), temperature, pressure, and DO, are shown in Table 2. Fatigue loading was applied using a strain-controlled method. The dimensions of the specimens used in the LCF test are presented in Figure 3. The fatigue life, $N_{25}$, is the number of cycles at which tensile stress during the fatigue cycle decreased to 75% of peak stress value, was used to evaluate the fatigue behavior of the tested materials. Consequently, the LCF tests were terminated when the tensile stress decreased by 25% of the peak stress value. After tests, the fatigue surface was observed to obtain the morphologies of fatigue crack using a scanning electron microscope (SEM) (SEM, JEOL, JSM-6480, Takeno, Japan).

**Table 1.** Mechanical properties of the Z3CN20.09M ASS at RT and 350 °C.

| Properties | Temperature | Properties | Z3CN20.09M |
|---|---|---|---|
| Tensile Properties | 25 °C | $R_{p0.2}$<br>$R_m$<br>A% (5d) | 235 MPa<br>655 MPa<br>56.8 % |
| | 350 °C | $R_{tp0.2}$<br>$R_m$ | 155 MPa<br>395 MPa |
| KV Impact | 25 °C | Lowest Average Value | 265 J |

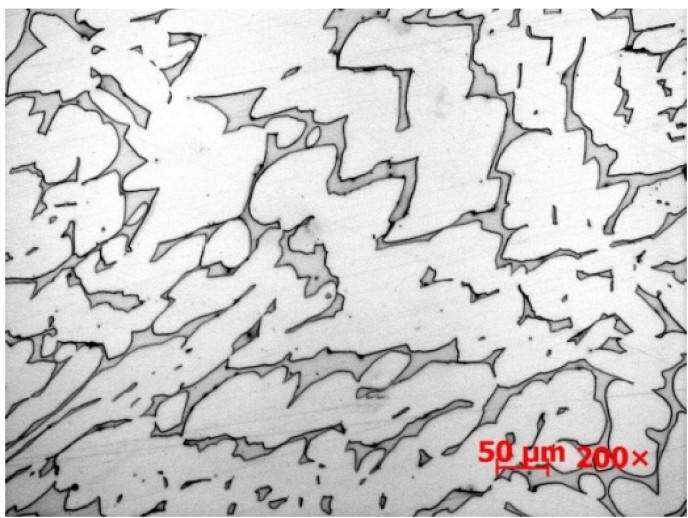

**Figure 2.** Optical observation of the Z3CN20.09M ASS.

**Table 2.** LCF test conditions and water chemistry.

|  |  |  |
| --- | --- | --- |
| | Load ratio (R) | −1 |
| | Control mode | Strain |
| Test conditions | Wave form | Full reversed triangular |
| | Strain rate | 0.4, 0.04 and 0.004%/s |
| | Strain amplitude ($\varepsilon_a$) | 0.2–1.0% |
| | Temperature | 300 °C |
| Water chemistry | Pressure | 12 MPa |
| | Conductivity | <0.1 μS/cm |
| | Dissolved oxygen (DO) | 10 ± 1 ppb |

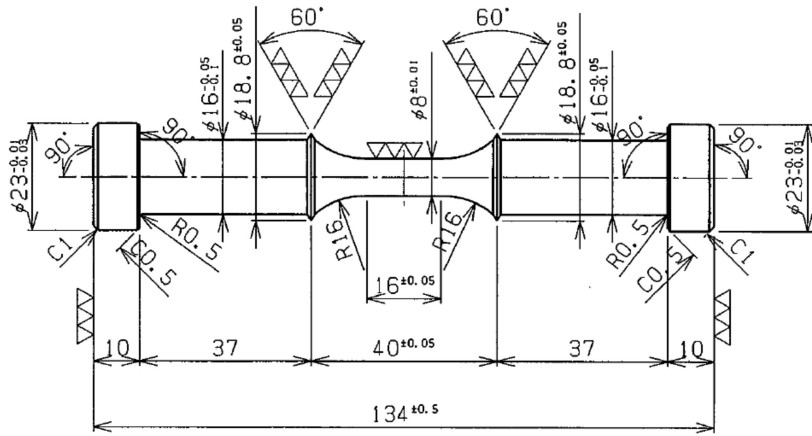

**Figure 3.** Schematic diagram showing the dimensions of the fatigue specimen.

## 3. Results and Discussion

### 3.1. Fatigue Life

The fatigue life, $N_{25}$, of the Z3CN20.09M austenitic SS in water at 300 °C is plotted against strain and shown in Figure 4. In Figure 4, the ASME design fatigue curve and the fatigue life in air at RT are also presented [4,8]. As is seen, in the same loading condition, the fatigue life of the Z3CN20.09M ASS in water at 300 °C was shorter than that in air at RT [2,3,6,9,13–15]. It indicates that the involvement of corrosive medium or the interactions of corrosive medium with applied load might be responsible for the decrease in the fatigue

life of the Z3CN20.09M ASS [16,17]. At the same strain amplitude, the decrease in strain rate yielded a lower fatigue life [15,18–21]. For example, at 0.8% strain amplitude, the decrease in strain rate from 0.04% to 0.004%/s led to 3.5 times lower in the fatigue life.

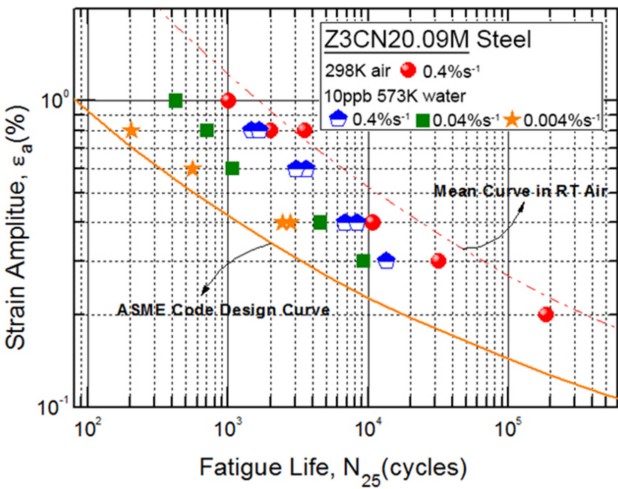

**Figure 4.** Fatigue life of the Z3CN20.09M ASS in water at 300 °C.

The plot of fatigue life of the Z3CN20.09M ASS in water at 300 °C as a function of strain rate is shown in Figure 5. Generally, the fatigue life of the Z3CN20.09M ASS in water at 300 °C decreased with the decreasing strain rate. Furthermore, the difference in the fatigue life at these three strain rates became more pronounced at higher strain amplitude. It might be caused by the higher corrosion rate of the tested material at higher strain amplitude in the autoclave environments [6,13–27].

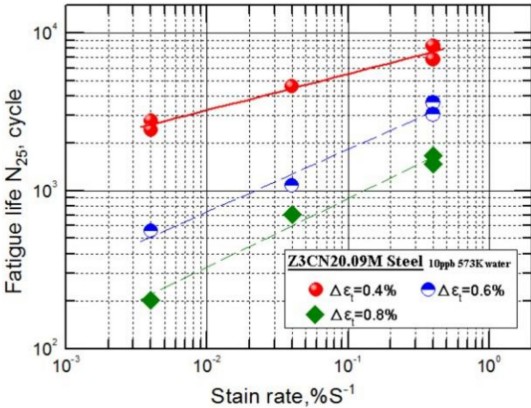

**Figure 5.** The plot of the fatigue life of the Z3CN20.09M ASS against strain rate in water. The testing corrosive medium was water, and the temperature was 300 °C.

### 3.2. SEM Observation

To explore the contribution of corrosive medium to the decrease in the fatigue life of the Z3CN20.09M ASS in water at 300 °C, the morphologies of fatigue surfaces at various strain rates were observed, Figure 6. As a comparison, Figure 6 also shows the fractographic surface of the Z3CN20.09M ASS specimen tested in air at RT. It is clear that well-developed ductile striations were present on the fatigue surface of the Z3CN20.09M ASS specimen when exposed to air at RT [12,27,28], Figure 6a. When the Z3CN20.09M ASS specimen was tested in water at 300 °C, its fatigue crack features were different from that in air at RT [4,6,22,23], as shown in Figure 6b,c. Additionally, the characteristics of the fatigue crack morphologies of the specimens tested in high temperature water changed with strain rates [13,22,23]. At a higher strain rate (i.e., 0.4%/s), several flattened regions marked using white dish lines in Figure 6b were observed, where many striations having "streamed

down features" were visible. At a lower strain rate [24] (i.e., 0.004%/s), these ductile fatigue striations with streamed down features were dominant at the fracture surface [13], Figure 6c. The presence of these striations with streamed down features is believed to be correlated with the occurrence of metal dissolution during the LCF test [18,25,29,30]. The metal dissolution process was proposed to occur via the following steps: (i) the formation of the oxide film at the crack tip by the anodic metal dissolution, which passivated the crack tip [23], (ii) the rupture of the passive film by applied strain, (iii) the exposure of fresh metal to the corrosive environment, leading to the reactivation, (iv) the dissolution of the freshly exposed metal [24], and (v) the propagation of fatigue crack and the increase in crack growth rate due to the interaction of the metal dissolution and applied load [31,32]. It means that the interaction of metal dissolution and applied load accelerated the crack propagation, leading to the decrease in the fatigue life of the Z3CN20.09M ASS in the simulated LWR water.

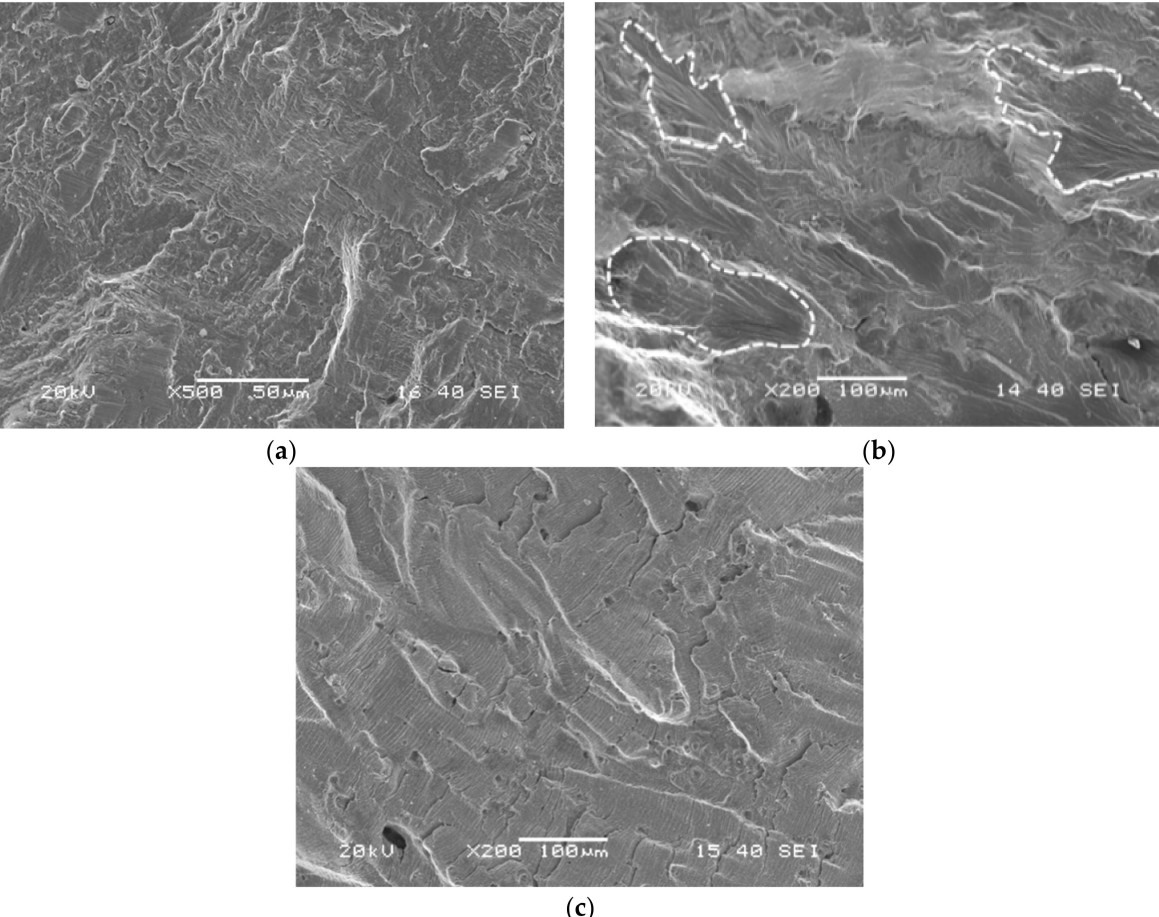

**Figure 6.** Fractured surfaces of Z3CN20.09M ASS at the strain amplitude of 0.8% and the strain rate of: (**a**) 0.4%/s in air at RT, (**b**) 0.4%/s and (**c**) 0.004%/s in high temperature water.

Figure 7 presents the cracks at the gauge sections of the LCF specimens tested in RT air and in simulated LWR water. At the same strain amplitude and strain rate, the crack propagation path in high temperature water was more tortuous than that in air, Figure 7a,d, suggesting that the corrosion medium was involved in the crack growth. It is consistent with the results of figure life measurements in Figure 5. With the increase of strain rate from 0.004%/s to 0.4%/s, the crack propagation path changed from being perpendicular to the loading direction to being zigzagged, Figure 7b–d.

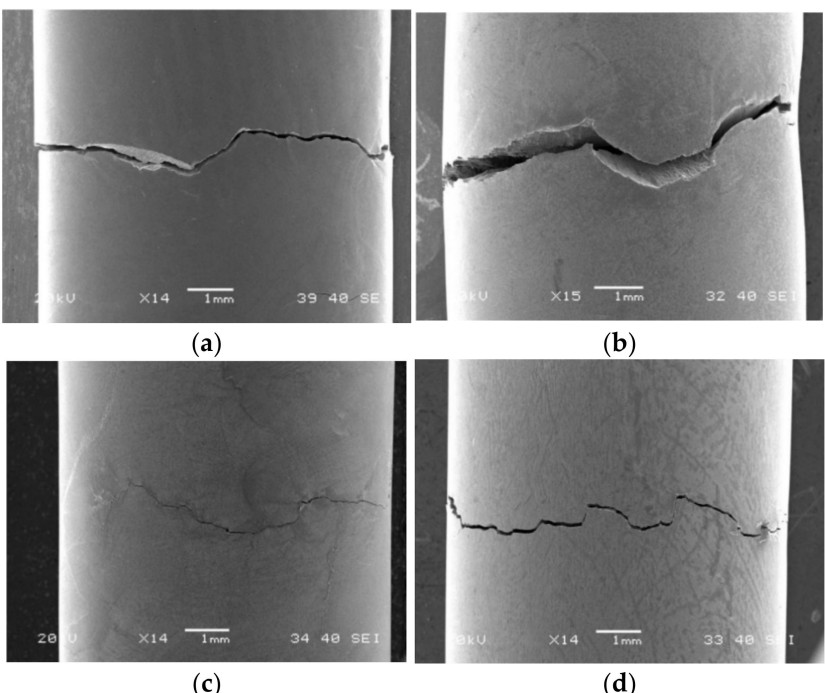

**Figure 7.** Surface crack morphology of LCF specimens tested at the strain amplitude of 0.8% and the strain rate of: (**a**) 0.4%/s in air at RT, (**b**) 0.004%/s, (**c**) 0.04%/s and (**d**) 0.4%/s in high temperature water.

The surface morphology at the crack tip of the LCF specimens after testing in high temperature water is shown in Figure 8. It can be seen that the crack tip had the features of interconnecting (Figure 8a), crossing (Figure 8b) and bifurcation [24] (Figure 8c,d). At the crack tip when the stress concentrated, many dislocations initiated were slipping along the slip bands, raising the electrochemical activity of the dislocation slipping regions, and thus accelerating their corrosion processes. It led to the accumulation of corrosion products at the crack tip (Figure 9) and the dislocation slipping regions, which resulted in the blunting of the crack tip. The cyclic application of tensile and compressive load would rupture the corrosion products formed at the crack tip and reactivate the corrosion process by the exposure of the fresh metal surface, leading to the sharpening of the crack tip. These repeating blunting and sharpening processes would prompt the growth of fatigue crack. On the other hand, new slipping bands may generate as the crack propagated, which would favor the initiation of micro-cracks in the front of the main crack because of the applied load and active corrosion interactions [19,29]. Under the action of applied load, the main crack would connect these micro-cracks, possessing the features of coalescing (Figure 8a), crossing (Figure 8b) and bifurcating (Figure 8c). This depended on the orientation of micro-cracks with the main crack.

### 3.3. Difference between the Fatigue Life Data and the Prediction Models

To ensure the reliability of the experimental data, two prediction models, ANL's model [3] and Higuchi's model [5] were used in the present work. The ANL's statistical model was established based on the Langer equation, in which several terms were added to consider the environmental factors influencing the fatigue life of SS [11,13,33,34]. In LWR environments, this model to predict the fatigue life of austenitic SS can be expressed using Equation (1) [3]:

$$\ln(N_{25}) = 6.157 - 192\ln(\varepsilon_a - 0.112) + T^* \cdot \varepsilon^* \cdot O^* \tag{1}$$

where $T^*$, $\varepsilon^*$, and $O^*$ is the transformed temperature, strain rate, and DO, respectively. These parameters can be determined by the test conditions in the present work.

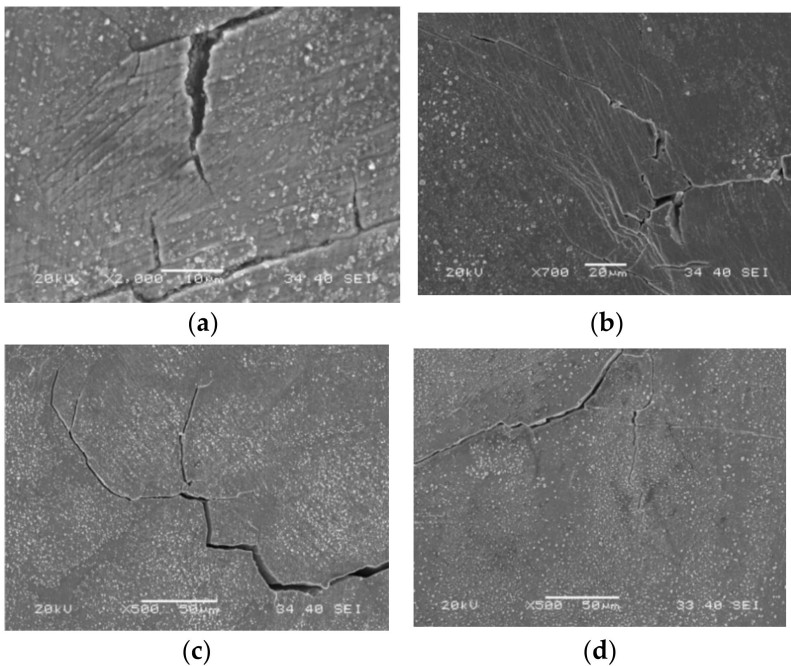

**Figure 8.** Crack tip morphology of the LCF specimens at the strain amplitude of 0.8% and the strain rate of 0.4%/s in high temperature water: (**a**) crack coalescing; (**b**) crack crossing; (**c**,**d**) crack bifurcating.

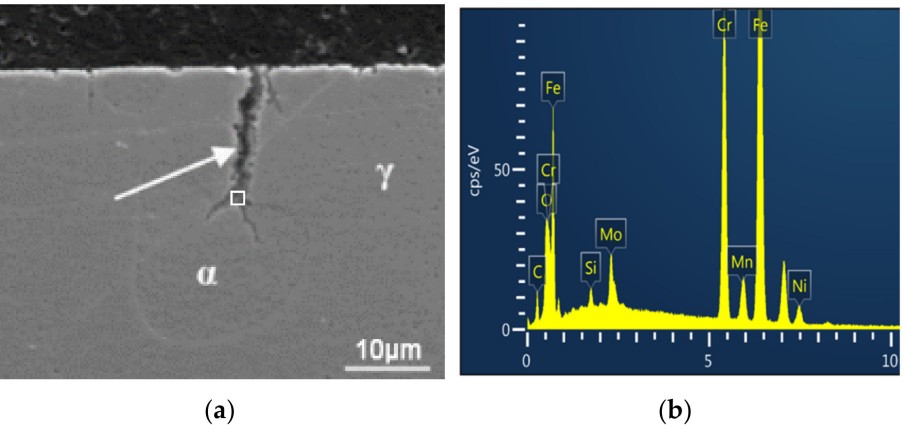

**Figure 9.** Cross-section morphology of secondary cracks (**a**) and the EDS analysis of corrosion products accumulated at the crack (**b**).

According to Higuchi et al. [5,10], the fatigue life correction factor, Fen, was proposed and defined using Equation (2):

$$F_{en} = \exp((C - \varepsilon^*)T^*)A^* \tag{2}$$

where, $C$ is a constant determined by the reactor type, and $\varepsilon^*$ *is* strain rate, $T^*$ *is* temperature and $A^*$ is strain amplitude. The fatigue life predicted using Higuchi's model was calculated by multiplying the fitted fatigue life of SS in air [9] with $F_{en}$ calculated from our test conditions [7,35,36].

The fatigue life obtained in this work and the predicted ones available using ANL's model and Higuchi's model are plotted in Figure 10. Generally, our experimental results are identical with the predicted fatigue life for these three strain rates [3,35,37]. Although the difference in the experimental data from the predicted one was within a factor of 3, it was considered to be acceptable. Thus, the fatigue life obtained in this work was acceptable

and reliable. However, it is necessary to further analyze the variability of data, which is an arduous and important task.

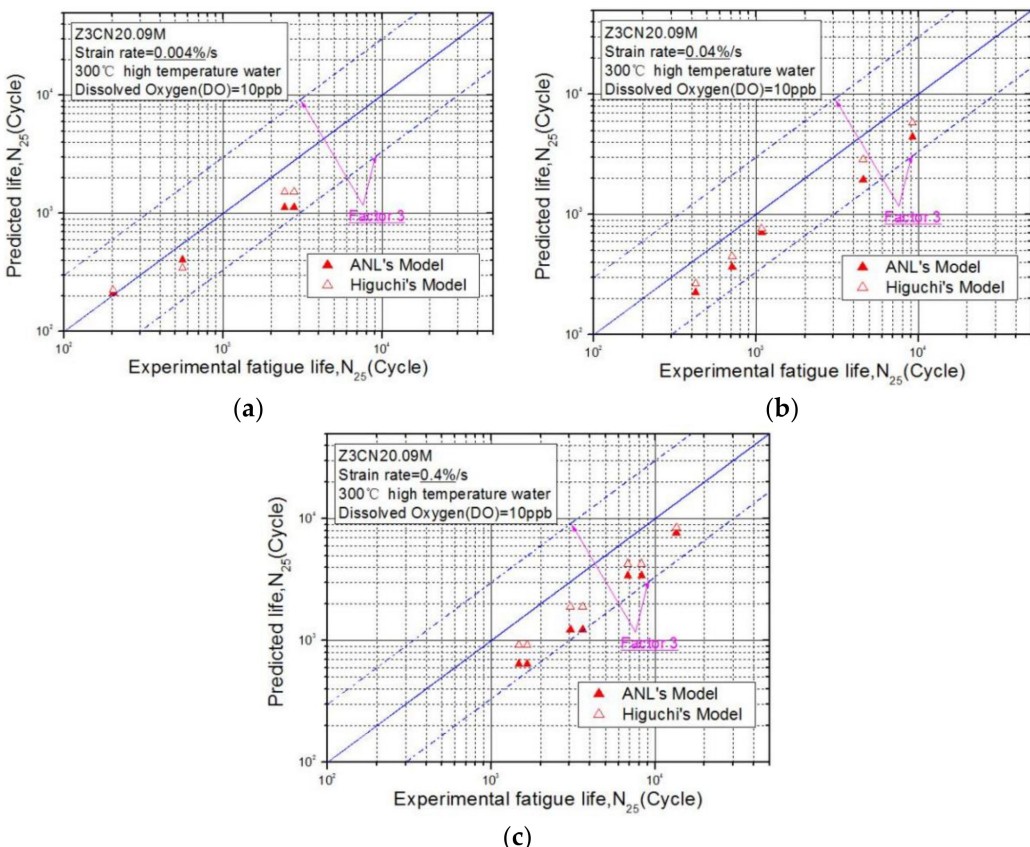

**Figure 10.** The fatigue life data obtained in the current work to the predicted fatigue life at the strain rate of: (**a**) 0.004%/s, (**b**) 0.04%/s and (**c**) 0.4%/s.

## 4. Conclusions

The LCF tests of the Z3CN20.09M austenitic SS were performed to investigate the fatigue life in high temperature water. Its fatigue life in 300 °C water was shorter than that in air under present testing conditions depending on strain rate, suggesting the involvement of corrosive medium. When the strain rate decreased from 0.4%/ to 0.004%/s, the reduction in the fatigue life became remarkable. The fractographic observation suggested that the decrease in fatigue life was attributable to the interaction between metal dissolution and applied load. Under the action of applied load, corrosion preferentially occurred at the places where dislocations initiated and accumulated, leading to the deposition of corrosion products. Owing to the expansion stress of corrosion deposit formation and applied stress, stress could easily concentrate at the bottom of corrosion damage, leading to the initiation of crack and the acceleration of the corrosion process. It in turn accelerated the accumulation of corrosion products at the crack tip and the dislocation slipping regions and thus, resulted in the blunting of the crack tip. Additionally, the cyclic application of tensile and compressive load would rupture the corrosion deposits to expose fresh metal surface and reactivate the corrosion process, leading to the sharpening of the crack tip. These repeating blunting and sharpening processes would facilitate the propagation of fatigue crack.

The comparison of the fatigue life obtained experimentally to the one predicted using ANL's model and Higuchi's model confirmed that our experimental data was reliable, and their differences were within the acceptable range.



**Author Contributions:** Conceptualization, C.L. and Y.Q.; methodology, K.L.; formal analysis, K.F.; investigation, K.F.; data curation, K.F.; writing—original draft preparation, K.F., C.L., and L.W. (Li Wang); writing—review and editing, L.W. (Lei Wang) and Y.Q.; supervision, Y.Q.; funding acquisition, K.F. and C.L. All authors have read and agreed to the published version of the manuscript.

**Funding:** This research received no external funding.

**Institutional Review Board Statement:** Not applicable.

**Informed Consent Statement:** Not applicable.

**Data Availability Statement:** The data used to support the findings of this study are available from the corresponding author upon request.

**Acknowledgments:** This work was jointly supported by the National Key Research and Development Program of China under (No. 2016YFB0700404), the Natural Science Foundation of China (No. 51375182) and the Major Natural Science Foundation of China under (No. U1260201).

**Conflicts of Interest:** The authors declare no conflict of interest.

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
