# Peer review of "Environmental Fatigue Behavior of a Z3CN20.09M Stainless Steel in High Temperature Water"

_coatings, doi:10.3390/coatings12030317_

Round 1

Reviewer 1 Report

This manuscript is interesting and appears to be useful but this reviewer has some issues. First, there is a partial superposition with a recent article and a much more complete article on Coating (Xue et al. Coatings 2021, 11, 870. https://doi.org/10.3390/). The manuscript appears to point to new things and can be considered complementary, but the other one must be cited. Another thing is the concept of “high temperature”. 300K can be considered “hot” or “warm” but not “high”.  So, my advice is to change this across the manuscript. In addition, a one-temperature study can be useful but is somehow limited.

The conclusions are trivial, as for corrosion water and oxygen are of utmost importance, and of course, hot water is more “corrosive” than air. Much more interesting conclusions must be quantitative. The authors do not attempt to fit their results but only to compare with standard models. 

The brand of stainless steel that the authors use appears to be standard, and the properties appear to be well known. The results in Table 1 appear to be from the authors. It is? (note the authors use the notations Z3CN20.09M and Z3CN20-09M, this must be harmonized).

It is not clear how “cycles” transform in time of use or how the strain rates are performed (The authors show results for 0,4%/s, 0,04%/s, and 0,0004%/s, and based on this, the time do not appear to be great). Again the work appears to be limited.

There are some misprints as “fa-tigue”, “nu-clear”, “vari-ability” due to the breack of pages. This and other possible ones must be corrected. I noticed also some possible issues with the equations (please look carefully). 

Also, it appears to be  acknowledgments something that must be also in funding.

In summary, the manuscript appears to be interesting for this journal but the work seems too limited and trivial to publish. My advice is to make more work and present more quantitative and sound conclusions to be more useful, explaining more.

Author Response

Reviewer 1

  1. This manuscript is interesting and appears to be useful but this reviewer has some issues. First, there is a partial superposition with a recent article and a much more complete article on Coating (Xue et al. Coatings 2021, 11, 870. https://doi.org/3390/). The manuscript appears to point to new things and can be considered complementary, but the other one must be cited.

Thanks for this comment. The paper the reviewer mentioned has been cited in the revised manuscript.

  1. Another thing is the concept of “high temperature”. 300K can be considered “hot” or “warm” but not “high”.  So, my advice is to change this across the manuscript. In addition, a one-temperature study can be useful but is somehow limited.

When the paper was submitted, the unit of temperature used in present work was not properly displayed, which was actually °C (300 °C) rather than K. This has been changed through the manuscript with a proper format to fit the requirements of the journal. In present work, the experimental temperature of 300 °C is locating in the range of high temperature normally termed in nuclear power field [2, 5, 6, 9].

  1. The conclusions are trivial, as for corrosion water and oxygen are of utmost importance, and of course, hot water is more “corrosive” than air. Much more interesting conclusions must be quantitative. The authors do not attempt to fit their results but only to compare with standard models. 

According to the reviewer’s comments, this section has been revised. We agree with the reviewer that our goal is to fit the results when the contribution of other factors (i.e., the influences of temperature, chloride concentration, dissolved oxygen and hydrogen) to the fatigue life of Z3CN20.09M is clear, which will be more reliable. Presently, these works are underway.

  1. The brand of stainless steel that the authors use appears to be standard, and the properties appear to be well known. The results in Table 1 appear to be from the authors. It is? (note the authors use the notations Z3CN20.09M and Z3CN20-09M, this must be harmonized).

The mechanical properties of studied material rely on the supply source and the testing environments, which is necessary to be investigated to determine whether the studied material fits the requirements of nuclear grade materials. In Table 1, its mechanical properties at 350 °C were also studied and provided as a comparison. The notation of the material should be Z3CN20.09M SS, which has been corrected in the revised manuscript.

  1. It is not clear how “cycles” transform in time of use or how the strain rates are performed (The authors show results for 0,4%/s, 0,04%/s, and 0,0004%/s, and based on this, the time do not appear to be great). Again the work appears to be limited.

The loading parameters have been given in Table 2 in detail. According to the obtained fatigue life in Figure 4 and 5, the test time was not short, which was dependent on the test loading parameter. As responded above, more extensive work such as the influences of temperature, chloride concentration and dissolved oxygen and hydrogen are on-going.

  1. There are some misprints as “fa-tigue”, “nu-clear”, “vari-ability” due to the breack of pages. This and other possible ones must be corrected. I noticed also some possible issues with the equations (please look carefully). 

These have been modified throughout the manuscript as the reviewer pointed out.

  1. Also, it appears to be acknowledgments something that must be also in funding.

It has been checked carefully and cited properly. Thanks to remind this.

Reviewer 2 Report

Grammatical correction to be done:

Page 1 Line 11

Page 1 Line 21

Page 1 Line 22

Page 4 Line 80

Page 4 Line 81

Page 5 Line 112

Page 5 Line 118

Others

Page 6 Line 134 to 135: Can you prove the accumulation of corrosion products at the crack tip by experimental evidence?

Page 6 Line 136 to 137: Can you provide the microstructural evidence of the rupturing of the corrosion products formed at the crack tip?  

Author Response

Reviewer 2

  1. Grammatical correction to be done:
  2. Page 1 Line 11
  3. Page 1 Line 21
  4. Page 1 Line 22
  5. Page 4 Line 80
  6. Page 4 Line 81
  7. Page 5 Line 112
  8. Page 5 Line 118

The grammar issues have been fixed. Thanks to catch them.

Others

  1. Page 6 Line 134 to 135: Can you prove the accumulation of corrosion products at the crack tip by experimental evidence?

Thanks for this comment. As shown below, it is quite clear that corrosion products analyzed using EDS accumulated at the crack tip. This piece of evidence has been included in the revised manuscript.

  1. Page 6 Line 136 to 137: Can you provide the microstructural evidence of the rupturing of the corrosion products formed at the crack tip?  

Thanks for this comment. Actually, we tried our best to find the evidence to proof the rupturing process of the corrosion products. It is quite hard to achieve this since the environments (e.g., temperature) changed after the autoclave was cooled down, and the structure and morphologies of the corrosion products might change when the autoclave was opened. But based on the present work and literature [19,29], the corrosion products had to be ruptured as the crack propagated, making the continuous accumulation of corrosion products at the crack tip and sustaining the sharpening of crack tip.

Round 2

Reviewer 1 Report

This version appears better, though the authors do not paint everything that they modified.